# The temporal regulation of TEK contributes to pollen wall exine patterning

**Shuang-Xi Xiong**[1☯], **Qiu-Ye Zeng**[2☯], **Jian-Qiao Hou**[2], **Ling-Li Hou**[2], **Jun Zhu**[2], **Min Yang**[2], **Zhong-Nan Yang**[2], **Yue Lou**[2]*

**1** School of Environmental and Geographical Sciences, Shanghai Normal University, Shanghai, China,
**2** Shanghai Key Laboratory of Plant Molecular Sciences, College of Life Sciences, Shanghai Normal University, Shanghai, China

☯ These authors contributed equally to this work.
* louyue5@shnu.edu.cn

**Data Availability Statement:** All relevant data are within the manuscript and its Supporting Information files.

**Funding:** This work was supported by the grants from "Chen Guang" project supported by Shanghai

## Abstract

Pollen wall consists of several complex layers which form elaborate species-specific patterns. In Arabidopsis, the transcription factor ABORTED MICROSPORE (AMS) is a master regulator of exine formation, and another transcription factor, TRANSPOSABLE ELEMENT SILENCING VIA AT-HOOK (TEK), specifies formation of the nexine layer. However, knowledge regarding the temporal regulatory roles of TEK in pollen wall development is limited. Here, TEK-GFP driven by the *AMS* promoter was prematurely expressed in the tapetal nuclei, leading to complete male sterility in the *pAMS:TEK-GFP* (*pat*) transgenic lines with the wild-type background. Cytological observations in the *pat* anthers showed impaired callose synthesis and aberrant exine patterning. *CALLOSE SYNTHASE5* (*CalS5*) is required for callose synthesis, and expression of *CalS5* in *pat* plants was significantly reduced. We demonstrated that TEK negatively regulates *CalS5* expression after the tetrad stage in wild-type anthers and further discovered that premature TEK-GFP in *pat* directly represses *CalS5* expression through histone modification. Our findings show that TEK flexibly mediates its different functions via different temporal regulation, revealing that the temporal regulation of TEK is essential for exine patterning. Moreover, the result that the repression of *CalS5* by TEK after the tetrad stage coincides with the timing of callose wall dissolution suggests that tapetum utilizes temporal regulation of genes to stop callose wall synthesis, which, together with the activation of callase activity, achieves microspore release and pollen wall patterning.

## Author summary

To develop into mature pollen grains, microspores require formation of the pollen wall. To date, pollen wall developmental events, including production and transportation of pollen wall components, synthesis and degradation of the callose wall, and deposition and demixing of primexine, have been studied in Arabidopsis, and a number of anther- or tapetum-specific genes involved in pollen wall formation have been uncovered. However, whether the specific expression patterns of these genes contribute to pollen wall

Municipal Education Commission and Shanghai Education Development Foundation (15CG50) to YL, the National Science Foundation of China (31600243) to YL and the Innovation Program of Shanghai Municipal Education Commission (2017-01-07-00-02-E00039) to JZ. The funders had no role in study design, data collection and analysis, decision to publish, or preparation of the manuscript.

**Competing interests:** The authors have declared that no competing interests exist.

development or patterning remains unclear. Here, we show that TEK, a transcription factor that specifies formation of nexine (the inner layer of the pollen wall exine), represses the expression of the callose synthase *CalS5* after the tetrad stage, which accurately fits with the timing of callose wall dissolution causing microspore release. Moreover, we show that premature expression of TEK in the wild-type anthers disturbs callose wall synthesis and pollen wall patterning. This work reveals that a pollen wall regulator must be kept under a strict temporal control to perform its functions, and that these temporal controls are coordinated with other pollen wall developmental events to determine pollen wall formation and patterning.

## Introduction

In angiosperms, the male gametophyte (pollen) is surrounded by a pollen wall. The pollen wall usually comprises two main layers, the outer sporophyte-derived exine and the inner gametophyte-derived intine. Distinctive pollen wall patterns vary between species but are conserved within species. In Arabidopsis, the exine creates a reticulate pattern on the pollen surface, containing layers known as sexine and nexine. The major constituent of sexine is sporopollenin, which is the biopolymer of polyhydroxylated aliphatic chains and aromatic rings [1–5]. Nexine includes arabinogalactan proteins (AGPs) [6], which are hydroxyproline-rich glycoproteins [7]. In turn, intine consists of cellulose, hemicellulose, pectin and proteins [8]. These features and structural components allow pollen wall to protect pollen from environmental stresses, such as desiccation, UV radiation and microbial attack, as well as allow it to provide species-specific adhesion to stigma [9–11].

The cellular events involved in pollen wall development have been most thoroughly studied in Arabidopsis [9, 10, 12–20]. The development of the pollen wall initiates in the individual microspores of tetrads after male meiosis. The callose wall first covers the plasma membrane of microspores. The domains where the plasma membrane closely contacts the callose wall are required for the formation of apertures. Later, the primexine matrix follows the undulation of the plasma membrane to guide the siting of sporopollenin. In the early tetrad stage, the callose wall, primexine, and plasma membrane participate in the formation of the sexine template. At the late tetrad stage, nexine starts to gradually accumulate in the primexine matrix. After callose wall degradation and microspore release, nexine completes its formation beneath sexine. Following the expansion of microspores, intine forms below the nexine, and pollen coat fills in the exine cavities. Therefore, these pollen wall components precisely appear at specific time points and sequentially assemble to form the elaborate pollen wall.

Pollen wall biosynthesis is a joint effort on the parts of microspores and the tapetum. The tapetum is the innermost sporophytic cell layer adjacent to microsporocytes and microspores [21]. It plays essential roles in pollen wall development because most pollen wall materials, including sporopollenin, lipids and proteins, are produced by, stored in, and transported from the tapetum [22]. Several tapetum-expressed genes are key players in directing the highly sculptured pollen wall in Arabidopsis and are involved in exine patterning [23–28], tapetum development and exine formation [29–34], sporopollenin biosynthesis [35–38], and nexine formation [6, 39]. Mutations in these genes lead to abnormal pollen walls and compromised pollen grains. Recent studies have shown that regulatory networks in the tapetum control pollen wall formation [40–44]. Among them, ABORTED MICROSPORES (AMS), a bHLH transcription factor, directly regulates the expression of *TRANSPOSABLE ELEMENT SILENCING VIA AT-HOOK* (*TEK*) [39]. Subsequently, *TEK*, encoding an AT-hook nuclear matrix

attachment region (MAR) binding protein, becomes highly expressed in the tapetum to promote nexine formation [39]. The investigations of mutants in which these genes are disrupted have demonstrated that these molecular players are required for the biosynthesis of multilayered pollen walls. However, whether the specific temporal patterns of these regulators contribute to pollen wall development or patterning remains unknown.

Because the *AMS* transcript is expressed earlier in the tapetum than the *TEK* transcript (anther stage 5 vs. stage 7, respectively) [39, 45], in order to explore whether the alteration in the temporal pattern of *TEK* disturbs pollen wall development, we used the *AMS* promoter to prematurely drive *TEK* expression. In the Arabidopsis *pAMS*:*TEK-GFP* transgenic plants with a wild-type background (hereafter referred to *as pat*), TEK-GFP was precociously expressed in the tapetal nuclei. We found that *pat* lines exhibited total male sterility and formed defective pollen walls. Ultrastructural observations showed that the callose wall formation was severely blocked and that the abnormal exine pattern existed in all *pat* microspores. Expression analysis confirmed that expression of the callose synthase *CalS5* is strongly decreased in *pat*, which could account for the reduced callose synthesis. We showed that, in wild-type anthers, TEK negatively regulates *CalS5* expression after the tetrad stage, which coincides with the callose wall degradation. Additionally, we found that the precocious TEK-GFP in *pat* directly represses the expression of *CalS5* via H3K9me2. These results showed that the correct regulation of timing of TEK expression is required for pollen wall exine patterning and is coordinated with other pollen developmental events for the processes of pollen wall formation and pollen grain maturation.

## Results

### Premature tapetum-specific expression of TEK-GFP protein in *pat*

To test whether premature expression of TEK can influence pollen wall development, we fused the *AMS* promoter with the *TEK-GFP* chimeric gene and introduced this construct (*pAMS*: *TEK-GFP*) into the wild-type Arabidopsis Columbia-0 (Col) plants (Fig 1A). Of the 26 transgenic plants, 21 plants exhibited male sterility and 5 plants were fertile. All male-sterile lines carried the transgene, as assessed by the amplification of the chimeric fragment, while fertile plants did not show amplified products (S1A and S1B Fig). Among these male-sterile plants, *TEK* expression was examined in three independent lines (*pat-1*, *pat-2* and *pat-3*). Quantitative RT-PCR (qRT-PCR) analysis showed that the total transcript levels of *TEK* in these lines were higher than those in the wild type (S1C Fig). Additionally, other cytological observations were performed in the *pat-1*, *pat-2* and *pat-3* lines, and the results were similar to each other (S2 and S3 Figs), leading us to uniformly refer to these lines as *pat*.

To determine whether this transgene leads to the ectopic expression of *TEK* in *pat* lines, we investigated its spatiotemporal patterns in anthers from wild-type and *pat* plants. Fourteen distinct stages of anther development have been described in Arabidopsis [21]. RNA *in situ* hybridization showed that *TEK* was expressed in the tapetum of the wild-type anthers throughout stages 7–8 (Fig 1B–1E), with the strongest hybridization signal at stage 7 (Fig 1D), in agreement with a previous study [39]. In the *pat* anthers, however, *TEK* transcripts were prematurely expressed during stages 5–6, with a strong signal in the tapetum and a relatively weak signal in the microsporocytes (Fig 1G and 1H). From stage 7 to stage 8, the hybridization signal in the tapetum peaked and later decreased after the release of microspores (Fig 1I and 1J), which was the same expression pattern as in the wild type (Fig 1D and 1E). No signal was produced in either wild-type or *pat* anthers with the sense probe (Fig 1F and 1K). These results indicated that *TEK* is expressed earlier in *pat* anthers than in wild-type plants. Additionally,

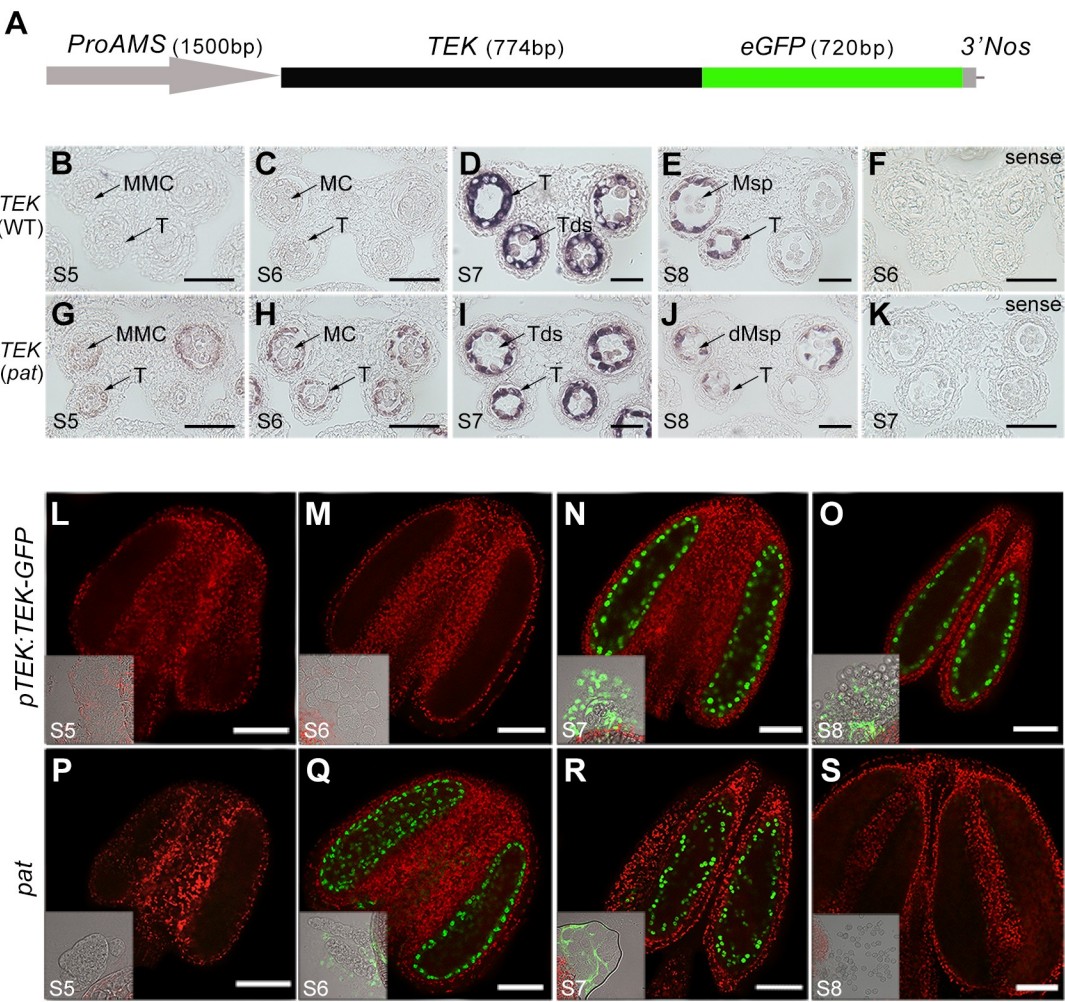

**Fig 1. Precocious expression of *TEK* transcripts and TEK-GFP proteins driven by the *AMS* promoter in *pat*.** (A) The *pAMS*:*TEK–GFP* constructs included a 1500 bp *AMS* promoter, *TEK* genomic fragment and GFP coding region. RNA *in situ* hybridization of *TEK* transcript in WT (B–E) and *pat* (G–J) anthers at stages 5–8 was performed using an antisense probe. *TEK* transcript in WT anthers at stage 6 (F) and *pat* anthers at stage 7 (K) using a sense probe. MMC, microspore mother cell; MC, meiocytes; T, tapetum; Tds, tetrads; Msp, microspore. Scale bars, 20 μm. Fluorescence confocal images of the TEK–GFP fusion protein in the anthers of *pTEK*:*TEK-GFP* transgenic plants (L-O) and *pat* (P-S) at stages 5–8. TEK-GFP was specifically located in the tapetal nuclei and expressed at stages 7–8 in *pTEK*:*TEK-GFP* transgenic anthers (N and O), while in *pat* anthers this protein was precociously expressed at stages 6–7 (Q and R). The bright-field images are located at the bottom left, showing that GFP fluorescence was observed only in the tapetal cells. Scale bars, 50 μm.

*AMS* expression level and pattern were not affected in *pat* anthers, excluding the possibility that male sterility was caused by perturbations in *AMS* expression (S4 Fig).

To further monitor TEK subcellular localization, we placed the *TEK-GFP* chimeric gene under its native promoter, and the construct *pTEK*:*TEK-GFP* was introduced into wild-type plants. In the *pTEK*:*TEK-GFP* transgenic plants, no fluorescence of TEK-GFP was displayed in the microsporocytes and tapetum during stages 5–6 (Fig 1L and 1M). Then, TEK-GFP specifically exhibited fluorescence in the nuclei of tapetal cells during stages 7–8 (Fig 1N and 1O). In the *pat* transgenic plants, the fluorescence of TEK-GFP was similarly restricted to the nuclei of tapetal cells but was prematurely expressed at stage 6 (Fig 1P and 1Q). The expression signal of

TEK-GFP was retained during the tetrad stage (stage 7) (Fig 1R), but only a very weak signal was observed in the tapetum at stage 8 (Fig 1S). These results demonstrated that TEK-GFP protein driven by the *AMS* promoter is precociously expressed in the tapetal nuclei of *pat* anthers.

## Male sterility is associated with microspore degeneration in *pat* plants

Compared to the wild-type plants, the *pat* transformants exhibited normal vegetative growth but developed short siliques and were completely male sterile (Fig 2A and 2B). To examine pollen viability in the anthers, we performed Alexander's staining to distinguish the aborted pollen grains from mature ones. In wild-type anthers, the mature pollen grains were stained purple (Fig 2C), whereas in *pat* anthers, pollen remnants were stained green (Fig 2D), suggesting that the pollen grains were aborted in *pat*. Anther cross-sections further showed that there was no detectable difference between wild-type and *pat* anthers until stage 6 (Fig 2G and 2L). Compared with the wild type, the callose of tetrads in *pat* seemed to be reduced at stage 7 (Fig 2H and 2M), which was confirmed by aniline blue staining (Fig 2Q and 2R). At stage 8, the

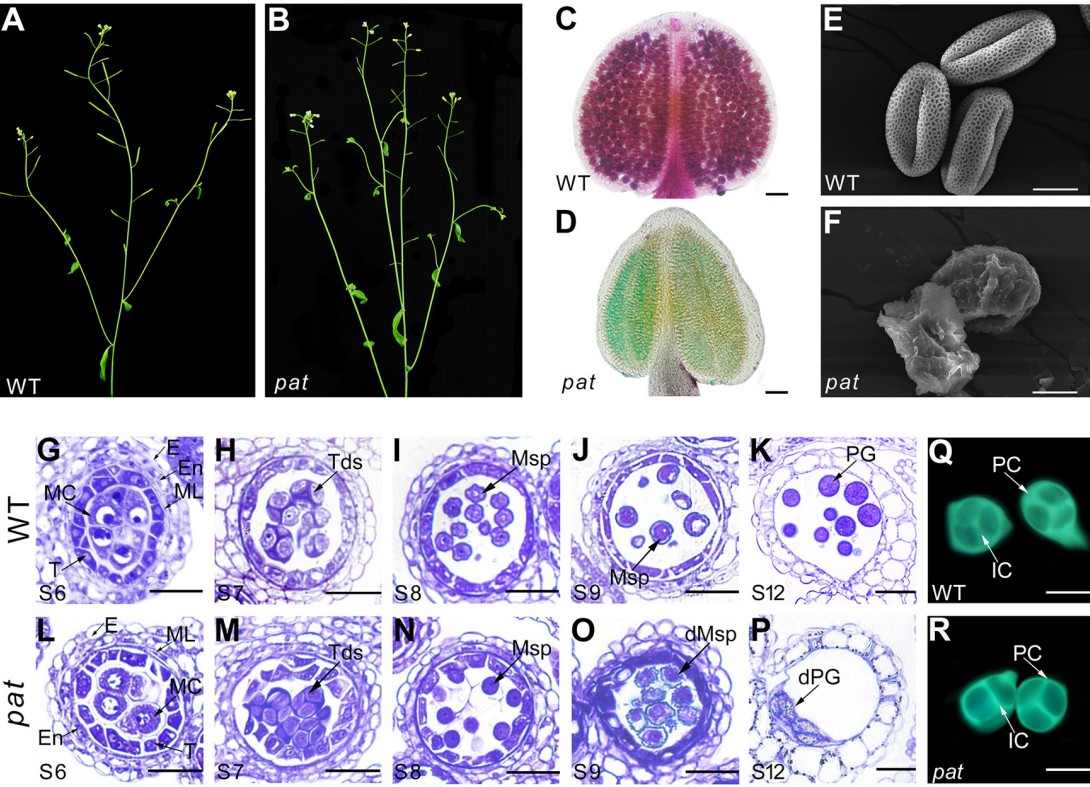

**Fig 2. Male sterility and microspore degeneration in *pat* plants.** (A) A wild-type (WT) plant has normal fertility. (B) A *pat* plant has short siliques and no seeds. (C-D) Alexander's staining of wild-type anthers (C), which contain viable pollen grains, stained purple, and *pat* anthers (D), which contain remnants of aborted pollen, stained green. Scale bars, 20 μm. (E-F) SEM observation of wild-type pollen grains (E) with reticulate pollen wall patterns and *pat* pollen grains (F) with irregular pollen wall patterns. Scale bars, 10 μm. (G-P) Semi-thin sections of WT and *pat* showing anther development during stages 6–12. In the wild-type anther (G-K), mature pollen grains were produced. In the *pat* anther (L-P), round microspores were released from tetrads (N) and gradually degenerated at later stages (O-P). E, epidermis; En, endothecium; ML, middle layer; T, tapetum; MC, meiocytes; Tds, tetrads; Msp, microspore; PG, pollen grain; dMsp, degenerated microspore; dPG, degenerated pollen grains. Scale bars, 5 μm. (Q-R) Aniline blue staining of callose from WT and *pat*. The peripheral callose wall of *pat* was slightly thinner (R) than that of the wild type (Q). PC, peripheral callose; IC, interstitial callose. Scale bars, 20 μm.

wild-type individual microspores with angular shapes were released from tetrads (Fig 2I). In contrast, the *pat* microspores were rounder than those of the wild type (Fig 2N). At stage 9, the wild-type microspores became vacuolated, while the *pat* microspores began to disintegrate (Fig 2J and 2O). At later stages, the wild-type microspores became noticeably enlarged, condensed their cytoplasm, and finally became mature pollen grains (Fig 2K). However, the *pat* microspores further degenerated, and only pollen remnants remained in the locule (Fig 2P). Scanning electron microscopy (SEM) showed the characteristic reticulate pattern on the surface of wild-type pollen, while a defective pollen wall was observed on the collapsed *pat* pollen (Fig 2E and 2F). These results showed that the failure of microspore production with disordered pollen wall morphology causes complete male sterility in *pat* plants.

## Defective exine development leads to microspore abortion in *pat*

It is acknowledged that aberrations in exine formation usually cause male sterility; therefore, we investigated the ultrastructure of the exine in *pat* plants by transmission electron microscopy (TEM). At stage 6, the wild-type microsporocytes underwent meiosis, and the callose wall accumulated around their periphery (Fig 3A and 3F). The accumulation of the callose wall in *pat* microspores was similar to that in wild-type microspores (Fig 3K and 3P). The tetrad stage is critical for exine formation [46, 47], as the stage during which the exine pattern is programmed through cooperation between the tapetum and microspores. In the wild type, primexine accumulated underneath the callose wall, and the plasma membrane became wavy in early stage 7 (Fig 3B and 3G). In contrast, in *pat* the callose wall was slightly sparse, the plasma membrane remained straight, and the sporopollenin precursors were located beside the primexine (Fig 3L and 3Q). In the middle of stage 7, the invagination of the wild-type plasma membrane was clearer, and the sporopollenin precursors from the tapetum deposited onto the peak to form the probacular within the primexine matrix (Fig 3C and 3H). In contrast, in *pat* the callose wall was thinner, and the sporopollenin precursors were randomly inserted into the primexine matrix (Fig 3M and 3R). At late stage 7, the probaculae elongated to the callose wall, and their distal ends fused into tectum, which constituted the pro-sexine in the wild type (Fig 3D and 3I). In *pat*, the sporopollenin precursors submerged into the primexine matrix, forming irregular shapes (Fig 3N and 3S). When the callose wall thoroughly dissolved at stage 8 in the wild type, nexine developed under the sexine, forming an intact exine with a T shape (Fig 3E and 3J). In *pat*, the irregular sporopollenin surrounded the microspores without any attachment (Fig 3O and 3T). Finally, the *pat* microspores degenerated because of the lack of pollen wall protection. These observations indicated that *pat* microspore abortion is due to abnormalities in exine development.

## Decreased *CalS5* expression leads to reduced callose synthesis in *pat*

Since the callose wall plays an important role in determining the exine pattern [48, 49], combined with the aberration of callose deposition in *pat* at the tetrad stage (Fig 3Q–3S), we explored the expression of some callose-related genes in *pat* inflorescences. *CalS5*, encoding a callose synthase (CalS), is required for callose synthesis around microsporocytes [48]. The mutations in *CalS5* affect the amount of callose deposition, leading to a disrupted exine pattern [48, 49]. *CYCLIN-DEPEDENT KINASE G1* (*CDKG1*) facilitates callose wall formation via the regulation of *CalS5* splicing [50]. *AUXIN RESPONSE FACTOR 17* (*ARF17*) regulates the expression of *CalS5* [43]. qRT-PCR analysis showed that only the *CalS5* transcript was greatly downregulated in the *pat* lines, while the others were not altered (Fig 4A). In addition, because primexine represents a decisive factor in exine ornamentation [24, 26, 27, 51], the expression of genes involved in primexine formation was also detected. We found that while the

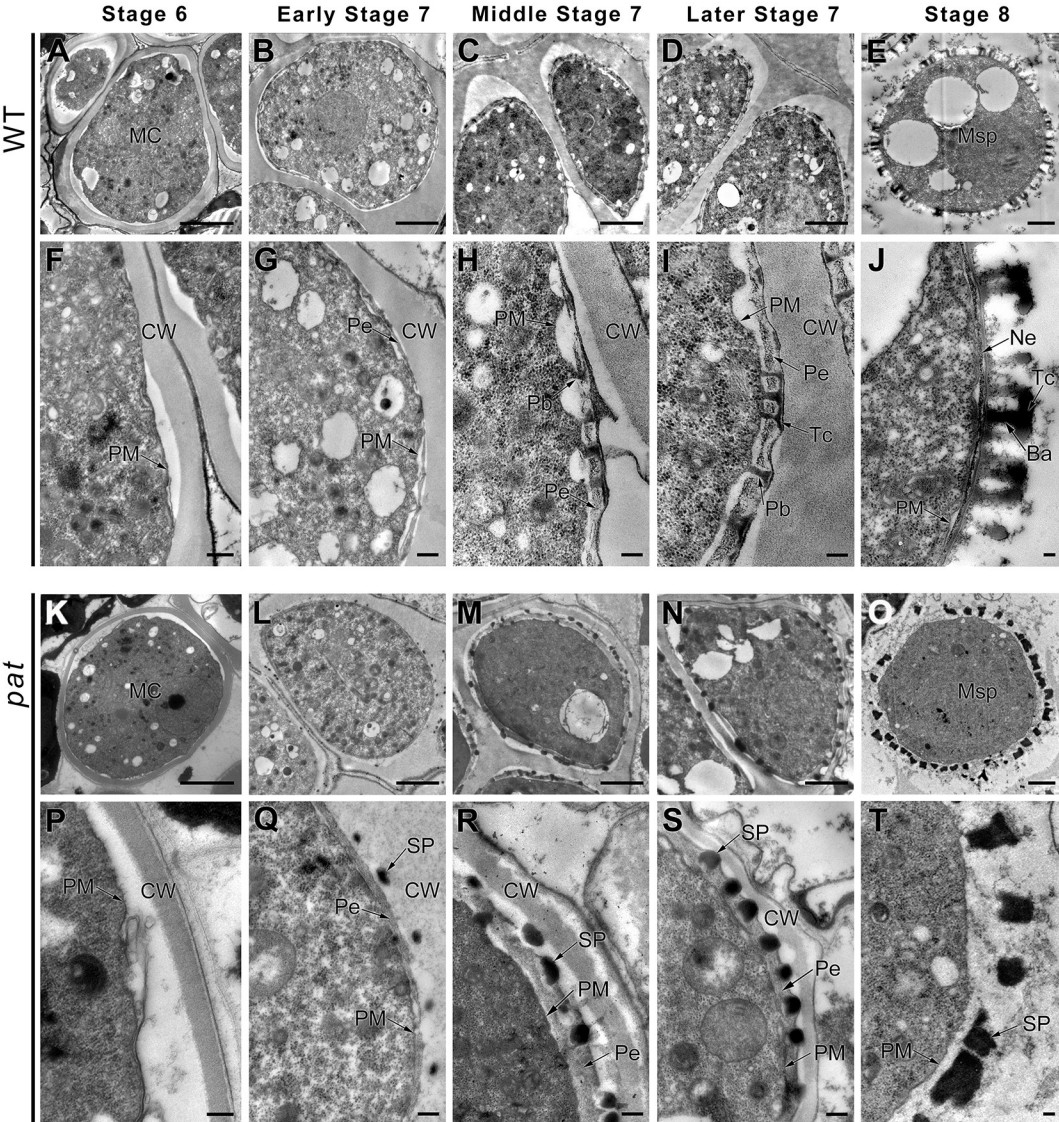

**Fig 3. Reduced callose deposition and defective exine development in *pat*.** TEM observation of pollen wall development in WT (A–E) and *pat* (K–O) at stages 6–8, magnified in (F-J) and (P-T), respectively. (P) Stage 6, showing that callose wall was deposited on the surface of *pat* microsporocytes. (Q) Early stage 7, showing that callose wall became slightly sparse and that sporopollenin precursors surrounded the primexine; (R) Middle stage 7, showing that the callose wall was thinner and sporopollenin precursors randomly inserted into the primexine matrix; (S) showing the irregular sporopollenin precursors submerged into the primexine matrix; (T) showing the defective exine patterning without attachment. CW, callose wall; MC, meiocytes; Pe, primexine; PM, plasma membrane; Pb, probacular; Tc, tectum; Ba, bacular; Ne, nexine; Msp, microspore; SP, sporopollenin precursors. Scale bars, 2 μm.

expression of *NO PRIMEXINE AND PLASMA MEMBRANE UNDULATION* (*NPU*) was slightly decreased, the levels of other genes were similar to those in wild type (Fig 4B), consistent with our TEM observation that primexine could still be formed in *pat* (Fig 3Q–3S).

To further confirm the reduction of callose synthesis in *pat*, we used aniline blue to stain callose and performed a callose fluorescence bleaching test. We randomly observed the tetrads from 12 independent tetrad-stage buds in the 12 independent WT and *pat* lines, recording the

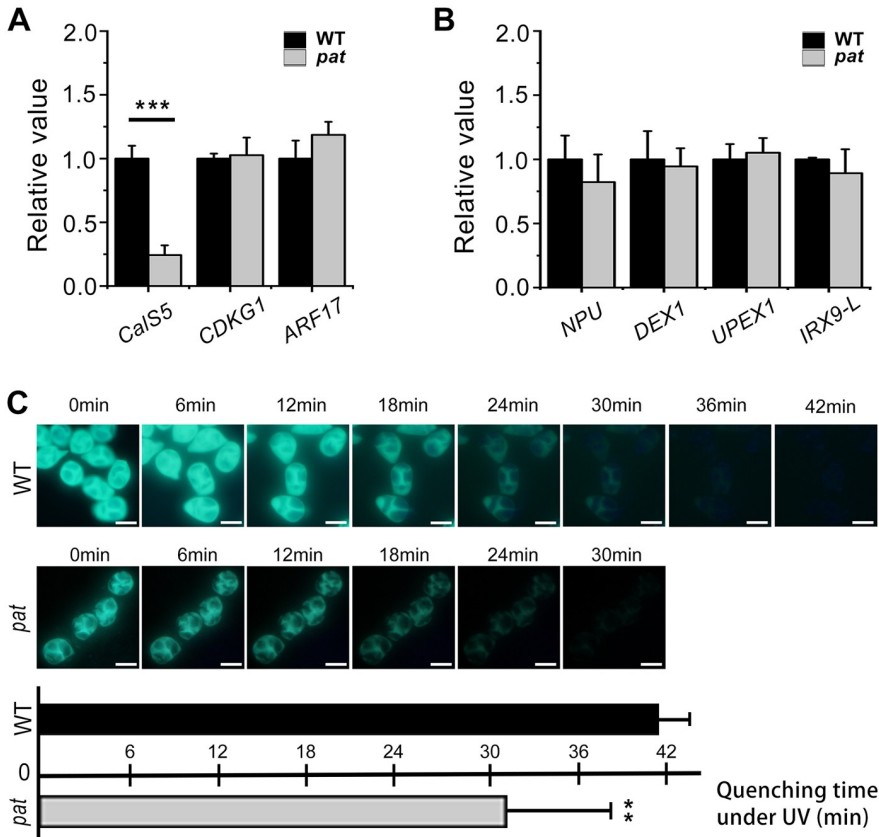

**Fig 4. Reduced callose synthesis in *pat* is associated with the decreased *CalS5* expression.** Expression of genes involved in callose synthesis (A) and genes involved in primexine formation (B) were detected in wild-type and *pat* plant inflorescences. Error bars represent the SD (n = 3). *** $p < 0.001$ (t-test). (C) Callose fluorescence quenching assay showed that callose wall fluorescence in *pat* quenched faster than that in WT. Error bars represent the SD of the mean of 12 biological replicates. ** $p < 0.01$ (t-test). Scale bars, 20 μm.

quenching time of each set (Fig 4C). In the wild type, the callose wall fluorescence of tetrads gradually quenched under UV. However, the callose wall fluorescence in *pat* quenched much faster. The average quenching time of callose wall fluorescence in *pat* was significantly less than in wild type. Therefore, these results showed that the defective callose synthesis in *pat* correlates with the severely reduced *CalS5* expression; this, in turn, may induce other defects in later pollen wall placement and patterning.

## Premature TEK-GFP controls *CalS5* expression via increasing H3K9me2 presence at the *CalS5* gene

Taken together, these results suggest that TEK-GFP proteins are prematurely expressed (Fig 1Q) and that the expression of *CalS5* is greatly decreased (Fig 4A). Is there a possibility that premature TEK-GFP affects the *CalS5* expression in *pat* anthers? To determine this, we investigated the *CalS5* expression pattern by *in situ* hybridization. In wild-type anthers, hybridization signals were predominantly observed in microsporocytes at stages 5–6 (Fig 5A and 5B). At stage 7, the signals were highly expressed in both the tapetum and tetrads (Fig 5C). No signal was found in the tapetum or microspores at stage 8 (Fig 5D). In the *pat* anthers, the signals

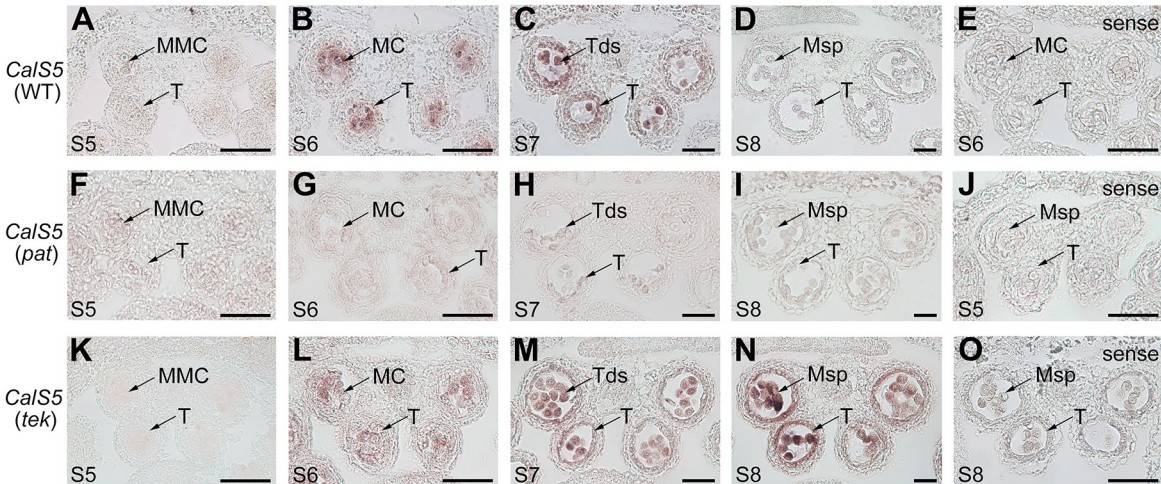

**Fig 5. *CalS5* expression patterns in the wild-type, *pat* and *tek* anthers.** Expression of *CalS5* in microspore mother cells, tetrads and tapetum was tested by RNA *in situ* hybridization in WT (A–D), *pat* (F-I) and *tek* anthers (K-N) at stages 5–8 using an antisense probe. According to the stages at which *CalS5* reaches its peak with an antisense probe, its transcript was observed with a sense probe in WT anthers at stage 6 (E), *pat* anthers at stage 5 (J), and *tek* anthers at stage 8 (O). MMC, microspore mother cell; MC, meiocytes; T, tapetum; Tds, tetrads; Msp, microspore. Scale bars, 20 μm.

at stage 5 showed no obvious difference from those in the wild-type anthers (Fig 5F). However, the signals in *pat* were reduced during stages 6–7 (Fig 5G and 5H and S5 Fig), consistent with the qRT-PCR data (S1C Fig). In *tek* anthers, the expression patterns of *CalS5* at early stages were similar to those in wild-type anthers (Fig 5K–5M). However, hybridization signals were still detected in both the tapetum and microspores at stage 8 (Fig 5N), suggesting that, in the wild-type, TEK specifically suppresses *CalS5* expression at this stage. Control hybridizations with the sense probe for *CalS5* did not show any signals in wild-type, *pat* and *tek* anthers (Fig 5E, 5J and 5O). Thus, considering that TEK-GFP was expressed precociously (Fig 1Q and 1R), we supposed that premature expression of TEK decreased *CalS5* expression in the *pat* anther.

As a putative MAR binding protein, TEK functions in silencing transposable elements (TEs) and repeat-containing genes by regulating histone dimethylation on H3K9 during flowering time regulation in *Arabidopsis* Landsberg *erecta*-0 (L*er*) [52]. We wondered whether this precocious TEK could repress *CalS5* expression via the similar mechanism in *pat* anthers. To test this hypothesis, we performed a chromatin immunoprecipitation (ChIP) assay using inflorescences from *pat* plants. It has been reported that MARs are AT-rich sequences of high affinity [53], and TEK binds MARs through the AT-hook motif [6]. We searched all putative MARs (A-box motif, WADAWAYAWW motif and AATATT motif) throughout the *CalS5* promoter and genomic sequence (Fig 6A). Primers were designed including or near the identified motifs to generate fragments of approximately 200 bp (Fig 6A and S6 Fig). Quantitative ChIP-PCR (qChIP-PCR) on putative TEK-GFP-binding sites showed that a genomic region of *CalS5* amplified by the primer set P11 was particularly enriched compared to the mock control (without the GFP monoclonal antibody) (Fig 6B). In contrast, a promoter region with the predicted MARs represented by primer set P3 showed no specific enrichment (Fig 6B). To further confirm the qChIP-PCR results *in vitro*, we performed an electrophoretic mobility shift assay (EMSA). First, the recombinant TEK protein fused to glutathione S-transferase (GST-TEK) was expressed in and purified from *Escherichia coli* (S7 Fig). Then, this GST-TEK protein and probes containing the P11 fragment (+2147 to +2332) were incubated together, which resulted

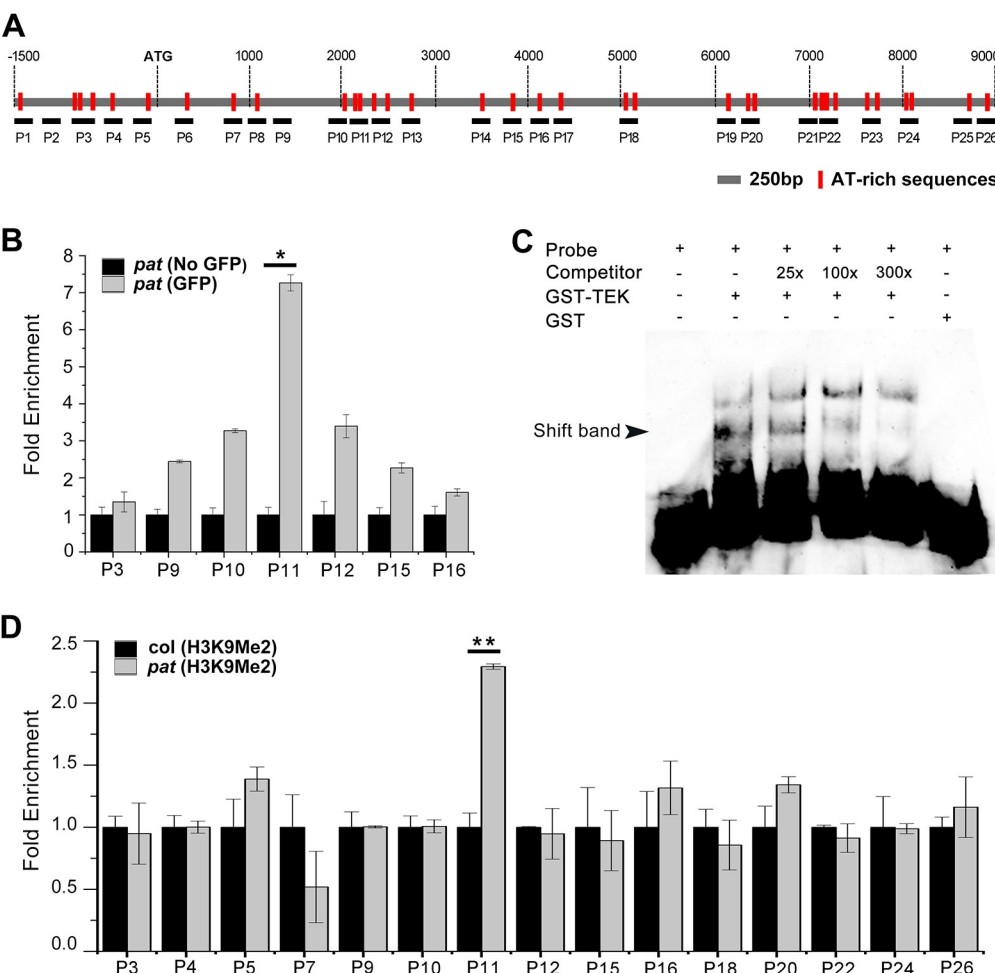

**Fig 6. Premature TEK-GFP binds to *CalS5* and increases H3K9me2 levels in *pat*.** (A) Schematic structure of the *CalS5* gene. AT-rich sequences present in the promoter and genome regions of *CalS5* are marked with red bars, and the corresponding primer pairs were designed to generate fragments of approximately 200 bp each. (B) qChIP-PCR results for several *Cals5* regions. ChIP was performed on *pat* inflorescences with (gray bars) or without (black bars) the GFP monoclonal antibody. Error bars represent SD (n = 2). * $p < 0.05$ (t-test). (C) EMSA assay was performed with the GST–TEK fusion protein, biotin-labeled probe that spanned the P11 fragment (+2147 to +2332), and a 25-fold, 100-fold and 300-fold excess of unlabeled competitor probes. Unlabeled competitors with the same P11 sequence are able to reduce the visible shift significantly (arrowhead). Glutathione S-transferase (GST) protein was expressed as a negative control. (D) ChIP performed on Col (black bars) or *pat* (gray bars) inflorescences with H3K9me2 antibody. Fold enrichment calculations from two replicate qPCR assays in three independent ChIP experiments. Error bars represent SD (n = 2). ** $p < 0.01$ (t-test).

in a specific band shift. No band shift appeared when only GST and the probe were co-incubated as a negative control. When the unlabeled probes were added, the intensity of shifted bands was gradually reduced in a concentration-dependent manner, confirming the binding specificity (Fig 6C). These results suggested that TEK directly binds to the specific MARs of *CalS5 in vivo*.

We then performed a ChIP assay to determine whether the repression of *CalS5* is associated with the change in H3K9me2. The inflorescences of wild-type and *pat* plants were collected, and antibody against H3K9me2 was used. qChIP-PCR showed that compared to the mock

control, H3K9me2 was specifically enriched in *pat* plants in the fragment of *CalS5* represented by the primer set P11 (Fig 6D and S2 Table). These results suggest that the premature appearance of TEK represses *CalS5* expression by modulating H3K9 dimethylation in *pat*.

## Discussion

To successfully grow into pollen grains, microspores require formation of the pollen wall. To date, multiple genes involved in formation of pollen wall layers have been identified and characterized [54]. However, whether the accurate timing of the expression of these genes is associated with pollen wall patterns and formation remains unknown. A previous study reported that the nexine layer is absent in the knockout mutant of *TEK* [39]. In the present study, we revealed that TEK specifically represses the expression of *CalS5* in wild-type anthers after the tetrad stage (Fig 5D and 5N). We further demonstrated that prematurely expressed TEK represses *CalS5* expression at the tetrad stage via histone modifications (Figs 1Q, 1R, 4A, 6B and 6D). This disturbs normal callose wall synthesis and exine patterning (Figs 2F, 3K–3T and 4C). Combined with previous studies, these results show that TEK not only activates the genes for nexine formation at the tetrad stage but also represses the expression of *CalS5* after the tetrad stage. Thus, these findings suggest that TEK, as a pollen wall regulator, executes its different functions in pollen wall patterning via differential temporal regulation.

In this study, defective callose synthesis resulting from reduced *CalS5* expression led to aberrant exine deposition (Fig 3K–3T). This result demonstrates that appropriate callose synthesis is essential for exine patterning during microspore development. Several theories on the biological functions of the callose wall have been advanced. Our TEM observation showed that primexine initially coupled to the contours of the plasma membrane and that the peaks of the undulating plasma membrane determined the sites for sporopollenin accumulation at the early tetrad stage. Meanwhile, almost no sporopollenin precursors derived from the tapetum were transported to primexine passing through the callose wall (Fig 3B and 3G). In contrast, in *pat* tetrads, the sporopollenin precursors penetrated through the callose wall and aggregated around the primexine (Fig 3L and 3Q). It is likely that the lower callose content in *pat* does not provide a sufficient barrier for sporopollenin precursors when the primexine matrix is not ready for their deposition. Therefore, in combination with the similar TEM data in *cdkg1* tetrads [50], these results seem to support the biological functions of the callose wall proposed by Heslop-Harrison (1964): that the callose wall acts as a chemical barrier to filter the molecules and isolate the haploid microspores in tetrads from the influence of the surrounding diploid tissues, ensuring normal pollen wall patterning [55].

Both primexine formation and membrane undulation are involved in pollen wall ornamentation [56]. Thin or absent primexine is usually found in callose deficient mutants, such as *cals5*, *cdkg1* and *arf17* mutants [43, 48, 50], suggesting that callose may provide a surface against which primexine is deposited or act as a source of glucose for primexine formation. In *pat* anthers, the primexine matrix still thickens normally at stage 7 (Fig 3Q–3S), and the expression of genes required for primexine formation is not affected (Fig 4B), suggesting that the reduced callose wall of *pat* may affect primexine for exine patterning through other routes. The primexine is described to be a polysaccharide material [15, 26], and the mixture of these polysaccharides is not stable [57]. Recently, it has been reported that polysaccharide materials tend toward demixing in the primexine, leading to spatially modulated phase separation. When primexine separation is in different states, it will form different templates for pollen wall deposition. It is speculated that membrane undulation in the vicinity of the callose wall induces this phase separation, and that components, including sporopollenin polymers and cellulose fibrils, arrest the phase separation [58]. We speculate that the reduced callose wall,

straight membrane, and early appearance of sporopollenin precursors in *pat* may alter the phase-separation process in the primexine, leading to a change in exine patterning.

During pollen wall development, microsporocytes and tapetum work in tandem. Initially, the callose wall is produced by microsporocytes during meiosis, and tetrads are encased by the callose wall, which marks the initiation of the exine pattern [59]. Subsequently, sporopollenin precursors produced from the tapetum are transported to specific places within the primexine [15], forming exine via self-assembly [60]. When the pro-sexine is formed, callase secreted from the tapetum dissolves the callose wall and releases microspores [61], marking the end of the exine patterning process. It has been reported that engineering callase activity to prematurely dissolve the callose wall produces microspores lacking normal pollen walls and causes male sterility in transgenic tobacco [62]. Therefore, the timing of callase secretion is critical for normal pollen wall development. In this study, we found that TEK, a tapetum-specific transcription factor, represses the expression of *CalS5* at stage 8 (Fig 5D), which coincides with the timing of callose wall degradation. This result suggests that tapetum not only provides the callase to dissolve the callose wall but also utilizes the temporal regulation of genes to stop callose wall synthesis, which together achieve microspore release and pollen wall patterning. In conclusion, we propose that the temporal control of pollen wall regulators exists in coordination with other processes to establish the intrinsic developmental timing scheme for whole pollen wall building and patterning.

## Materials and methods

### Plant materials and growth conditions

*Arabidopsis thaliana* accession Columbia (Col-0) was used for all gene transfer experiments and as wild-type controls. Plants were grown on soil in a growth room under long-day conditions (16 h light/8 h dark) at approximately 22–24˚C.

### Generation of constructs and transgenic plants

To generate the constructs *pAMS*:*TEK-GFP* and *pTEK*:*TEK-GFP*, a 771-bp genomic fragment of *TEK* was amplified from the wild type by KOD Neo Plus polymerase (Toyobo Co., Ltd., Osaka, Japan). The PCR product was cloned into a modified GFP-pCAMBIA1300 vector. Then, a 1500-bp *AMS* promoter and a 920-bp *TEK* promoter were amplified. These PCR products were individually digested by restriction enzymes (Takara Biotechnology) and ligated into the plasmid. After confirmation by restriction digestion and DNA sequencing, the resulting constructs were transformed into *Agrobacterium tumefaciens* GV3101, and the plants were transformed using the floral dip method [63]. The transformants were screened on Plant Nutritional Solution (PNS) media containing 20 mg/L hygromycin B and later transferred into the soil for PCR identification. Primer sequences are presented in S1 Table.

### Phenotype characterization

Alexander staining was performed as described [64]. All plants were photographed with a Nikon D700 digital camera. For cross-sections, flower buds from WT and *pat* were fixed overnight in FAA (ethanol 50% (v/v), acetic acid 5.0% (v/v), and formaldehyde 3.7% (v/v)), dehydrated in a graded ethanol series (50% [×2], 60%, 70%, 80%, 90%, 95%, and 100% [×2]), and embedded in resin with a low viscosity kit (PELCO, USA). Transverse sections of 1 μm in thickness were stained in 0.5% toluidine blue and observed with an Olympus BX51 microscope (Olympus, http://www.olympus-global.com).

## Expression analysis

Total RNA extraction was performed using TRIzol (Life Technologies) following the protocol in the user's manual. cDNAs of wild-type and *pat* inflorescences were used for the expression analysis of selected genes. Real-time quantitative PCR was performed using gene-specific primers and SYBR Green Master Mix (TOYOBO) on the ABI 7300 platform (Applied Biosystems). The experiments were repeated three times, and the data were averaged. The β-tubulin gene was used as an internal normalization control [65]. Fold changes in gene expression were calculated using the ΔΔCt (cycle threshold) values. The relevant primers are listed in S1 Table.

## Fluorescence microscopy

For callose staining, anthers at the tetrad stage were squeezed onto the slide and stained by aniline blue solution (0.1 g/L in 50 mM K3PO4 buffer, pH 7.5) [64]. Aniline blue was observed under UV illumination on an Olympus BX51 fluorescence microscope. The TEK-GFP localization in anthers of *pTEK*:*TEK-GFP* and *pat* lines was detected by a Carl Zeiss confocal laser scanning microscope (LSM 5 PASCAL; Zeiss, http://www.zeiss.com).

## RNA *in situ* hybridization

The probe fragment was amplified from the wild-type cDNA using primers (see S1 Table for primers). The PCR products were cloned into the pBluescript II SK (-) vector (Strata gene; http://www.stratagene.com) and confirmed by sequencing. Plasmid DNA was completely digested with EcoRI or BamHI. Antisense and sense digoxigenin-labeled probes were prepared using T3 or T7 RNA polymerase by the PCR DIG Probe Synthesis Kit (Roche, USA). Images were obtained with an Olympus BX-51 microscope. More details were described previously [45].

## Transmission electron microscopy (TEM) and scanning electron microscopy (SEM)

Arabidopsis buds from wild type and *pat* were fixed in 2.5% glutaraldehyde in 10 mM phosphate buffer (pH 7.4). Samples were post-fixed in 1% osmium tetroxide and dehydrated in an ethanol/water series (30, 50, 75, 85, 90, 95 and 100%). Then, the samples were dehydrated twice with 100% propylene oxide. Samples were subsequently transferred to 1:1, 1:3 and 3:1 propylene oxide/Spurr's resin mixtures and kept overnight. Later, samples were embedded in Spurr's resin and polymerized at 65°C for 48 h. Ultrathin sections (70-nm thick) were cut using diamond knives and stained in a solution of uranyl acetate and lead citrate. The images were viewed on a Hitachi H-600 transmission electron microscope (Hitachi Ltd, http://www.hitachi.com). A scanning electron microscopy assay was performed as described by [33].

## Chromatin immunoprecipitation assay

In the T1 generation, the *pat* transformants were confirmed by PCR identification and all of them showed male sterility. Then, WT pollen was used to cross-fertilize several independent *pat* stigmas to produce F1 seeds. In the F1 generation, the male-sterile plants were confirmed by PCR to have transgene insertion. Inflorescences from the male-sterile *pat* plants with transgene insertion were the materials used for the ChIP assay. A total of 0.8–1.0 g inflorescences from wild-type and *pat* plants were collected and crosslinked in the formaldehyde-containing buffer. After isolating the nuclei and shearing the chromatin with ultrasonication, most DNA fragments had a size between 200–800 bp. After preimmunization with sheared salmon sperm DNA/protein A agarose mix (Millipore, USA) for 1 h, supernatants were incubated at 4°C

overnight with monoclonal antibodies (1:125 dilution) against either GFP or dimethylated H3K9. Seventy microliters of magnetic beads coupled with protein G (Invitrogen) were added to precipitate the antibody–protein/DNA complexes. The DNA fragments were eluted after reverse crosslinking at 65°C overnight. The remaining steps for DNA purification were carried out according to the manufacturer's instructions. Real-time PCR was performed on an ABI PRISM 7300 detection system (Applied Biosystems, USA) with SYBR Green I master mix (TOYOBO, Japan). All PCR experiments were performed under the following conditions: 95°C for 5 min, 40 cycles of 95°C for 10 s and 60°C for 1 min. Under the same conditions, we calculated the ΔCt values and used $2^{-\Delta Ct}$ as the fold enrichment. The relevant primers are listed in S1 Table.

## Electrophoretic mobility shift assay

To obtain purified TEK protein, the full-length fragment of TEK was amplified and ligated into the pGEX-4T vector (GE Healthcare, http://www.gehealthcare.com) to generate the construct GST-TEK. Expression and purification of the fusion protein were performed according to the manufacturer's instructions. Corresponding primers (P11-F/P11-R) were used to amplify the probes with 5' biotin labeling. The competitor probes contained the same DNA sequence but lacked the 5' biotin labeling. EMSA was performed according to the manufacturer's instructions (Thermo Scientific, Waltham, MA, USA). Primer sequences are listed in S1 Table.

## Supporting information

**S1 Fig. Identification of independent *pat* transgenic plants.** (A) In the T1 generation, the presence of insertion in independent *pat* transgenic plants was confirmed by PCR. A 694-bp DNA including the *AMS* promoter and *TEK* genomic fragment was amplified using primers PAMSJD-F and CTEKJD-R. The plants without the insertion of the target fragment were fertile and were named FP (Fertile Plants). (B) Three independent *pat* transgenic lines are shown, and they are all male sterile, as confirmed by Alexander's staining of anthers. Scale bars, 20 μm. (C) Expression of *CalS5* and *TEK* was detected in three independent *pat* lines by qRT-PCR analysis. Error bars represent the SD (n = 3). *** $p < 0.001$ (t-test).
(TIF)

**S2 Fig. Expression analysis of independent *pat* transgenic plants.** RNA *in situ* hybridization of *TEK* transcripts in anthers of *pat-1* (A–D), *pat-2* (E–H) and *pat-3* (I–L) at stages 5–8 using an antisense probe. MMC, microspore mother cell; MC, meiocytes; T, tapetum; Tds, tetrads; dMsp, degenerated microspore. Scale bars, 20 μm. Fluorescence confocal images of the TEK–GFP fusion protein in anthers of *pat-1* (M-P), *pat-2* (Q-T) and *pat-3* (U-X) at stages 5–8. Scale bars, 50 μm.
(TIF)

**S3 Fig. Characterization of independent *pat* transgenic plants.** Semi-thin sections of *pat-1* (A-E), *pat-2* (F-J) and *pat-3* (K-O) showing anther development from stages 6–12. E, epidermis; En, endothecium; ML, middle layer; T, tapetum; MC, meiocytes; Tds, tetrads; Msp, microspore; dMsp, degenerated microspore; dPG, degenerated pollen grains. Scale bars, 5 μm. SEM observation of pollen grains in *pat-1* (P), *pat-2* (Q) and *pat-3* (R). Scale bars, 10 μm. The callose fluorescence quenching assay showed that callose wall fluorescence in *pat-1* (S), *pat-2* (T) and *pat-3* (U) quenched faster than that in WT (V). Scale bars, 20 μm. TEM observation of

tetrads in *pat-1* (W), *pat-2* (X) and *pat-3* (Y) at stage 7 compared with that in WT (Z). PC, peripheral callose. Scale bars, 2 μm.
(TIF)

**S4 Fig. Expression pattern of *AMS* in *pat*.** RNA *in situ* hybridization of *AMS* transcripts in the anthers of WT (A–D) and *pat-3* (F–I) at stages 5–8 using an antisense probe. *AMS* transcript in anthers of WT (E) and *pat-3* (J) using a sense probe at stage 6. MC, meiocytes; T, tapetum; Tds, tetrads; Msp, microspore; dMsp, degenerated microspore. Scale bars, 20 μm. (K) Expression of *AMS* was detected in three independent *pat* lines by qRT-PCR analysis. Error bars represent the SD (n = 3).
(TIF)

**S5 Fig. Expression pattern of *CalS5* in wild-type and independent *pat* anthers.** Expression of *CalS5* in microspore mother cells, tetrads and tapetum was detected by RNA *in situ* hybridization in anthers of WT (A–C), *pat-1* (E-G), *pat-2* (I-K) and *pat-3* (M-O) at stages 5–7 using an antisense probe. *CalS5* transcript in WT (D) and *pat* anthers (H, L, P) using a sense probe. MMC, microspore mother cell; MC, meiocytes; T, tapetum; Tds, tetrads. Scale bars, 20 μm.
(TIF)

**S6 Fig. Genomic sequence of *CalS5* with the positions of primers used for ChIP.** There are 26 pairs of primers for ChIP marked by blue serial numbers. The text highlighted in yellow indicates the AT-rich sequences. Underlined text indicates the detailed locations of primers.
(TIF)

**S7 Fig. SDS-PAGE analysis of recombinant TEK protein.** SDS-PAGE analysis of GST-TEK proteins used for *in vitro* EMSA analysis. Purified proteins were run on an 8% gradient gel and stained with Coomassie blue. M, protein markers.
(TIF)

**S1 Table. List of primers used in this research.**
(XLSX)

**S2 Table. qRT-PCR data of expression analysis and ChIP assay.**
(XLSX)

## Acknowledgments

We thank Xiao-Feng Xu, Cheng Zhang, Xiao-Zhen Yao and Hua Jiang for helpful discussion and revision.

## Author Contributions

**Conceptualization:** Shuang-Xi Xiong, Jun Zhu, Yue Lou.

**Formal analysis:** Shuang-Xi Xiong, Qiu-Ye Zeng.

**Funding acquisition:** Jun Zhu, Yue Lou.

**Investigation:** Shuang-Xi Xiong, Qiu-Ye Zeng, Jian-Qiao Hou, Ling-Li Hou, Min Yang.

**Methodology:** Shuang-Xi Xiong, Qiu-Ye Zeng, Yue Lou.

**Project administration:** Yue Lou.

**Resources:** Zhong-Nan Yang.

**Writing – original draft:** Yue Lou.

**Writing – review & editing:** Jun Zhu.

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
