## [Decision Letter · Decision Letter 0]

28 Oct 2019

Dear Dr Lou,

Thank you very much for submitting your Research Article entitled 'The temporal regulation of TEK contributes to pollen wall exine patterning' to PLOS Genetics. Your manuscript was fully evaluated at the editorial level and by independent peer reviewers. The reviewers appreciated the attention to an important problem, but raised some substantial concerns about the current manuscript. Based on the reviews, we will not be able to accept this version of the manuscript, but we would be willing to review again a much-revised version. We cannot, of course, promise publication at that time.

Should you decide to revise the manuscript for further consideration here, your revisions should address the specific points made by each reviewer and the guest editor. We will also require a detailed list of your responses to the review comments and a description of the changes you have made in the manuscript.

If you decide to revise the manuscript for further consideration at PLOS Genetics, please aim to resubmit within the next 60 days, unless it will take extra time to address the concerns of the reviewers, in which case we would appreciate an expected resubmission date by email to plosgenetics@plos.org.

[LINK]

We are sorry that we cannot be more positive about your manuscript at this stage. Please do not hesitate to contact us if you have any concerns or questions.

Yours sincerely,

Anna A. Dobritsa

Guest Editor

PLOS Genetics

Gregory Copenhaver

Editor-in-Chief

PLOS Genetics

**Guest editor's comments:**

Both reviewers conclude - and I agree with their assessment - that the findings are potentially interesting but you will need more evidence to demonstrate that male sterility in pat plants is indeed caused by the mechanism which you propose: that the premature expression of TEK in tapetum leads to problems with the formation of the callose wall around microspores and that TEK directly represses the callose synthase CalS5 gene by interacting with its regulatory regions and causing the appearance of repressive histone marks. As the analysis of a single transgenic line cannot be considered sufficient in this case (the effects could be caused by the position of the transgene insertion), you would need to perform the critical experiments which led to these conclusions on at least two more independent transgenic lines. This is particularly important because the mechanism that you propose is not consistent with the more accepted idea that microsporocytes themselves synthesize their callose walls. Therefore, you need to ensure that your conclusions, which imply that a lot of CalS5 activity comes from tapetum, are very strongly supported by experimental evidence.

In addition to providing detailed answers to the questions and concerns raised by the reviewers, I would encourage you to provide more evidence of the following:

1. The callose wall defects and the timing of their appearance in pat lines. The callose wall defects are far from obvious. There is little difference in the appearance of the callose wall in WT and in pat in the light-microscopy images (Figs. 1H and 1N, Figs. S2A and S2B). Even in TEM images (Figs. 3L and 3Q), for which you state that the “callose wall was obviously sparse”, this is not at all obvious and requires some strong quantitative support.

2. Timing of TEK expression has to be tested in additional pat lines.

3. Reduction of Cals5 expression has to be verified in additional pat lines, and I would suggest that this should be also demonstrated with methods other than just qRT-PCR.

4. The binding of TEK to the P11 region and the enrichment of H3K9Me2 in this region has to be tested in additional pat lines. Further, the ability of TEK to bind to this region has to be confirmed by an independent method, such as EMSA. Please also provide more information on the position of the P11 region in relation to the CalS5 gene model (e.g. Is it in an intron? In which one?) as well as the details for the qPCR experiments performed in conjunction with the ChIP experiments. Fig. 6A shows 26 regions that were supposedly tested by ChIP-qPCR, yet the graphs in Figs. 6B and C show the results from much fewer regions. What was done on the other regions? Also, to be confident in results of qPCR, they have to be performed very carefully – e.g. all primers have to be tested for their ability to double the amount of DNA at each cycle and Ct values have to be in a certain range to be reliable. Please provide evidence that all the proper controls were done for each tested region and the results were indeed reliable. Please add the supplemental table with all the primers (it seems to be missing from the submission).

5. Fig. 5 describes potentially important results but is presented so inconspicuously that even one of the reviewers missed the fact that the bottom row shows the CalS5 in situ hybridization in the tek mutant, not in pat. I would encourage you to also provide the in situ hybridization data on the Cals5 expression from several pat lines (and perhaps compare it with the CalS5 expression in the TEKpro:TEK-GFP lines) to strengthen your conclusions.

6. Please explain what is shown in the eight unlabeled lanes in Fig. S1A and provide the figure legend for Fig. S1C.

7. Remove the results on MS188 presented in Fig. S3. It is distracting when they are suddenly mentioned in the Discussion without being presented in the Results.

Reviewer's Responses to Questions

**Comments to the Authors:**

Reviewer #1: The Manuscript ‘The temporal regulation of TEK contributes to pollen wall exine patterning’ by Xiong et al. presents a series evidences to state the temporal expression of TEK is essential for pollen exine patterning. Previous studies have revealed that TEK is required for nexine formation and the tek mutant exhibited pollen nexine absence in Arabidosis. Here in this manuscript, the authors performed an earlier than normal expression of TEK (AMS::TEK-GFP, name as pat) in wild type anther and found the premature expression of TEK leads to completely male sterility; they employ cytological and molecular analyses to demonstrated that TEK physically interacts with the P11 site of CalS5 promoter and represses the CalS5 expression via H3K9me2.

Overall, this is an interesting story that provided a mechanism that how the tapetum stop the callose wall synthesis and promote the callose wall degeneration to achieve the microspores release and the pollen wall patterning. The technical quality of the study could however be improved. I summarize my comments and suggestions below:

1. I would suggest to do phenotypic analyses with more than two independent pat alleles, as least in some of the phenotypic analyses. Although fertility and alexander staining anther from 3 alleles were presented in S1, further cytological observations results were provided only in one allele.

2. On a more detailed level, the manuscript lack of experimental details. This is general, but I take several as example:

1) I could not find any indication of Figure S1C, in which relative expression of TEK in only one allele is shown.

2) We all know that different transgenic alleles could possibly cause developmental defects in different levels, as results of different expression level. Therefore, I would suggest the authors to show the relative expression of TEK in pat-1, pat-2, and pat-3, and clearly state the allele used in each following study but not only address ‘pat mutant’.

3) I could not find any indication of which statistical tests were used and which level of confidence are applied in Figure 4 and Figure 6.

3. Given the pat mutant showed reduced CalS5 expression and increased H3K9Me2 in CalS5 promoter, we would expect to see the increased CalS5 expression and decreased H3K9Me2 in tek mutant. I would invite the authors to test these to improve the precision and impact of these analyses.

Reviewer #2: The current work has its foundation in an earlier work by Lou et al (reference no 38 in the manuscript). The earlier paper had demonstrated the role of TEK gene in pollen development, wherein a knockout of the gene led to male sterility. This paper also showed that the TEK gene was directly regulated by AMS gene. In the current work the authors analyze the effects of expressing the TEK gene at stages of anther development earlier to that of its normal expression. This they achieve by expressing the TEK gene under the AMS promoter in transgenic Arabidopsis lines. Although the observations are interesting, the conclusions drawn from the observations seem improper.

My major concern is:

The authors report that under the AMS promoter the TEK gene is expressed precociously in stage 5 and 6 in the pat transgenic lines as is shown in Fig 1. An important observation that there seems to be reduction in TEK expression at stage 8 in the pat lines (compare Fig 1E to 1J) is not discussed. Is the pattern presented in the Figure observed in independent transgenic lines developed by the authors?

This quantitative observation is important to reach conclusions on the observed expression pattern of CalS5 gene in the pat transgenics (Fig 6). It is observed that the main difference in expression of Cals5, between WT and the pat lines lies in stage 8 of anther development. The expression of CalS5 is prominent in S8 in pat lines while there seems to be no expression in wild type (compare Fig 6D and 6I). There is no difference in expression between WT and pat lines in the earlier stages. The authors also record this observation (lines 241 to 246). However, they tend to argue that “Thus, considering that TEK-GFP expressed precociously, we supposed that premature TEK decreases CalS5 expression in pat anther”. The observation however, tends to show that the precocious expression of TEK in stage 5 and 6 does not change the expression pattern of CalS5, which they also mention in the text. However, the down regulation of TEK at stage 8 in pat lines leads to expression of CalS5 gene at this stage Thus, while TEK does seem to down regulate CalS5, the observations tend to show that it happens at a later stage (8) rather than at stages 5 and 6.

The results section concludes by saying “All these results suggested that premature appearance of TEK represses CalS5 expression via modulating the H3K9 demethylation in pat” (Line 271)

However, the initial part of discussion (line 281) the authors mention that “In the present study, we revealed that TEK specifically represses the expression of CalS5 in wild type anther after the tetrad stage (Fig 5I)”, which seems to be correct but not in line with the arguments given in the results section. The authors then discuss that the phenotypic outcome is because of the down regulation of Cals5 in the in the earlier stages which is not supported by the presented observations.

It is suggested that the authors reanalyze their observations and discuss it with more clarity.

Other points are:

i. Introduction on pollen development can be shortened. The introduction could concentrate more on the role of TEK in development of pollen, highlighting the initial study by Lou et al. The reason for asking the present question should also be mentioned in the introduction.

ii. As the TEK gene was driven by the promoter of the AMS gene, the spatial and temporal expression pattern of the AMS gene needs to be introduced. It would have been nice if AMS expression patterns had been shown in WT as well as in the pat transgenic lines, as perturbations in AMS expression can also lead to male sterility as has been observed in the earlier work. Thus it needs to be demonstrated that in the transgenic lines developed there is no changes in the AMS expression pattern.

iii. The manuscript mentions that because AMS is upstream to TEK (line 107), the expression of AMS should be at an earlier stage. A gene being upstream to another gene, does not ensure its expression at an earlier stage. It could express at the same stage but as a cascade.

iv. The details of the constructs used for developing transgenic lines should be presented. Line 130: It is not clear what is meant by lines that did not carry insertion: were they escapes during hygromycin selection or they carried the hptII but lacked the AMS:TEK transgene?

v. Line 138: In the wild type anther TEK expression is observed in S7 and S8 (both in situ and GFP in TEK:TEK-GFP lines) and not from S6-8 as mentioned in the text.

vi. How many independent transgenic lines were analyzed for spatial expression of the different genes by in situ hybridizations? Is the presented pattern observed in all the lines tested or only in a subset?

vii. The reduction in expression of the TEK gene at stage 8, in pat lines as compared to WT should be critically evaluated. The possible reasons for the same should be discussed.

viii. In experiments with qRT-PCR, the number of transgenic lines analyzed should be clearly mentioned. Currently it mentions that experiments were repeated three times. Are these replications from different RNA samples or three times from the same RNA sample.

ix. The evidence or literature showing that β-tubulin is an appropriate gene for normalizations in the stages studied should be mentioned.

x. The Ct values corresponding to the relative values presented in Fig. 4 could be included as a supplementary table. This helps in evaluating the strength of the presented data.

xi. In experiments of qRT-PCR or ChIP, it seems that an inflorescence has been used a sample. This would mean that buds of different stages would be present and thus conclusions from the observations become weak. It is thus important that results of at least three independent transgenic lines are presented to see the general trends.

**Have all data underlying the figures and results presented in the manuscript been provided?**

Reviewer #1: Yes

Reviewer #2: Yes

PLOS authors have the option to publish the peer review history of their article (what does this mean?). If published, this will include your full peer review and any attached files.

Reviewer #1: Yes: Fang Chang

Reviewer #2: Yes: Pradeep Kumar Burma

---

## [Decision Letter · Decision Letter 1]

3 Mar 2020

Dear Dr Lou,

Thank you very much for submitting your Research Article entitled 'The temporal regulation of TEK contributes to pollen wall exine patterning' to PLOS Genetics. Your manuscript was fully evaluated at the editorial level and by independent peer reviewers. The reviewers appreciated the attention to an important topic but identified some aspects of the manuscript that should be improved.

We therefore ask you to modify the manuscript according to the review recommendations before we can consider your manuscript for acceptance. Your revisions should address the specific points made by the reviewer 2.

Also, prior to resubmission, please pay special attention to ensuring that the grammar, punctuation, and spelling of your article are at a very high level because PLOS will not offer detailed copyediting in the event of eventual acceptance. Currently, the entire manuscript suffers from poor grammar. The main text, abstract, and author summary should all be easy to read and understand. We strongly recommend that you carefully review your paper with the assistance of a native/fluent English speaker or a professional language editing service before you submit a revised manuscript. We can, on request, offer the names of individuals with whom we have worked whom you could engage to assist you with your text. This would help to ensure optimal

quality and clarity of presentation within your revised manuscript.

[LINK]

Yours sincerely,

Anna A. Dobritsa

Guest Editor

PLOS Genetics

Gregory P. Copenhaver

Editor-in-Chief

PLOS Genetics

Reviewer's Responses to Questions

**Comments to the Authors:**

Reviewer #1: no more comments

Reviewer #2: Comments on the revised version:

i. The authors have presented results of three independent transgenic lines in the revised manuscript. Although the authors uniformly call the three independent lines as pat lines, I feel that they should be presented as pat-1, pat-2 and pat-3. Further, data from one of the lines say pat-1 is presented in the main text and those of the additional 2 lines in the supplementary.

ii. The gene that has been amplified to check the presence of transgene in supplementary figure 1, should be mentioned in the legends. Why does one of the WT plants in this figure have a similar amplification profile as the transgenic lines?

iii. While the text (line 103) mentions that AMS transcripts are observed in anthers from stage 5, supplementary figure 4 shows results from stage 6 only. If possible the expression at stage 5 should also be included.

iv. I feel lines 126 to 128 should be rewritten as: "Of the 26 transgenic lines, 21 were observed to be male sterile and 5 lines were fertile. All male sterile lines carried the transgene as assessed by amplification of (mention the gene that was amplified) fragment while fertile events did not show amplified product." The lines marked as WT in the amplification profile in supplementary figure 1 are actually not untransformed wild type plants but those that probably were male fertile and also lacked the amplification product. This needs to be clearly mentioned.

v. Lines 129 to 134 should be written with clarity, taking into consideration the first point.

vi. The section on CalS-5 expression in wild type, tek mutant lines and pat transgenic lines should be clearly written. Figure 5 should also include the hybridization profile of one of the pat lines. This will help a reader get a comparative view and understand the conclusion easily.

vii. In the gel retardation assay, the meaning of unlabeled competitor probe is not clear. Is it unlabeled P11 fragment? If yes, then one cannot make any conclusions on biding specificity. Further, the conclusions are drawn based on the second band (from top), without commenting on the band which represents maximum retardation. With the observed % of fragments that are retardation, one would have expected no retardation in 300 fold excess.

**Have all data underlying the figures and results presented in the manuscript been provided?**

Reviewer #1: Yes

Reviewer #2: Yes

PLOS authors have the option to publish the peer review history of their article (what does this mean?). If published, this will include your full peer review and any attached files.

Reviewer #1: Yes: FANG CHANG

Reviewer #2: Yes: pradeep kumar burma

---

## [Editor Report · Decision Letter 2]

23 Apr 2020

Dear Dr Lou,

Thank you very much for submitting your Research Article entitled 'The temporal regulation of TEK contributes to pollen wall exine patterning' to PLOS Genetics. Your manuscript was fully evaluated at the editorial level and by independent peer reviewers. The reviewers appreciated the attention to an important topic.

The Editor has suggested several corrections to the language of the manuscript, which you will find in the attached document. We ask that you please consider this document and make the appropriate changes to the text of your manuscript, at which point we will be prepared to accept the manuscript for publication.

Upload a Striking Image with a corresponding caption to accompany your manuscript if one is available (either a new image or an existing one from within your manuscript). If this image is judged to be suitable, it may be featured on our website. Images should ideally be high resolution, eye-catching, single panel square images. For examples, please browse our archive. If your image is from someone other than yourself, please ensure that the artist has read and agreed to the terms and conditions of the Creative Commons Attribution License. Note: we cannot publish copyrighted images.

[LINK]

Yours sincerely,

Anna A. Dobritsa

Guest Editor

PLOS Genetics

Gregory P. Copenhaver

Editor-in-Chief

PLOS Genetics

---

## [Editor Report · Decision Letter 3]

28 Apr 2020

Dear Dr Lou,

We are pleased to inform you that your manuscript entitled "The temporal regulation of TEK contributes to pollen wall exine patterning" has been editorially accepted for publication in PLOS Genetics. Congratulations!

Yours sincerely,

Anna A. Dobritsa

Guest Editor

PLOS Genetics

Gregory P. Copenhaver

Editor-in-Chief

PLOS Genetics

Comments from the reviewers (if applicable):

**Data Deposition**

http://datadryad.org/submit?journalID=pgenetics&manu=PGENETICS-D-19-01564R3

**Press Queries**

---

## [Editor Report · Acceptance letter]

7 May 2020

PGENETICS-D-19-01564R3 

The temporal regulation of TEK contributes to pollen wall exine patterning 

Dear Dr Lou, 

We are pleased to inform you that your manuscript entitled "The temporal regulation of TEK contributes to pollen wall exine patterning" has been formally accepted for publication in PLOS Genetics! Your manuscript is now with our production department and you will be notified of the publication date in due course.

With kind regards,

Matt Lyles

PLOS Genetics

On behalf of:
